# Development and Validation of a Predictive Nomogram for Venous Thromboembolism Risk in Multiple Myeloma Patients: A Single-Center Cohort Study in China

**DOI:** 10.3390/biomedicines13040770

**Published:** 2025-03-21

**Authors:** Haolin Zhang, Xi Zhang, Xiaosheng Li, Qianjie Xu, Yuliang Yuan, Zuhai Hu, Yulan Zhao, Yao Liu, Yunyun Zhang, Haike Lei

**Affiliations:** 1Chongqing Cancer Multi-Omics Big Data Application Engineering Research Center, Chongqing University Cancer Hospital, Chongqing 400030, China; zhl1281536698@163.com (H.Z.); xuqianjie22@163.com (Q.X.); cq_yuanyuliang@163.com (Y.Y.); 2Chongqing Key Laboratory of Translational Research for Cancer Metastasis and Individualized Treatment, Chongqing University Cancer Hospital, Chongqing 400030, China; zhangxi15223434456@163.com (X.Z.); toxiaosheng@163.com (X.L.); zhaoyulan1516@163.com (Y.Z.); liuyao77@cqu.edu.cn (Y.L.); 3Department of Health Statistics, School of Public Health, Chongqing Medical University, Chongqing 400016, China; 2022120774@stu.cqmu.edu.cn

**Keywords:** multiple myeloma, VTE, risk prediction, nomogram

## Abstract

**Objectives:** Venous thromboembolism (VTE) is a significant complication in patients with multiple myeloma (MM) that adversely affects morbidity, mortality, and treatment outcomes. This study aimed to develop and validate a predictive nomogram for assessing VTE risk in MM patients using clinicopathological factors. **Methods:** Clinical data, including 25 candidate risk factors, were collected. Univariate and multivariate logistic regression analyses were performed to identify independent risk factors for VTE. The nomogram was constructed using these variables, and its performance was evaluated by plotting receiver operating characteristic (ROC) curves, calculating the area under the curve (AUC), and conducting calibration and decision curve analysis (DCA). Additionally, an online calculator was developed for clinical use. **Results:** In total, 148 patients (17.5%) developed VTE in this study. The independent risk factors included age, Karnofsky performance status (KPS), anticoagulation therapy, erythropoietin use, and hemoglobin (Hb), platelet (PLT), calcium (Ca), activated partial thromboplastin time (APTT), and D-dimer levels. The nomogram demonstrated robust discriminative ability, with a C-index of 0.811 in the training cohort and 0.714 in the validation cohort. The calibration curves exhibited a high level of agreement between the predicted and observed probabilities. DCA confirmed the nomogram’s clinical utility across various threshold ranges, outperforming the “treat all” and “treat none” strategies. **Conclusions:** This study successfully developed and validated a nomogram for predicting VTE risk in MM patients, demonstrating substantial predictive accuracy and clinical applicability. The nomogram and accompanying online calculator provide valuable tools for individualized VTE risk assessment and informed clinical decision-making.

## 1. Introduction

Venous thromboembolism (VTE), which encompasses deep vein thrombosis (DVT) and pulmonary embolism (PE), is a critical condition caused by abnormal venous clot formation [1]. These clots can obstruct venous blood flow, either partially or entirely, leading to complications such as post-thrombotic syndrome and chronic thromboembolic pulmonary hypertension [2]. These complications significantly reduce quality of life for cancer patients and impose a heavy burden on healthcare systems, leading to frequent hospitalizations and treatment interruptions [3]. In multiple myeloma (MM) patients, VTE represents one of the most frequent and severe complications, with an incidence exceeding 10% [4]. More concerningly, MM patients with VTE face a threefold increase in mortality within a year compared with those without VTE [5]. Nonetheless, awareness of VTE risk factors among Chinese hematologists remains insufficient. Many physicians do not adequately implement preventive or therapeutic measures for VTE when choosing MM treatment regimens [6]. Furthermore, there is an absence of systematic risk stratification for VTE prevention and insufficient practical experience among clinicians. Given the profound impact of VTE on MM patients, including treatment delays, increased mortality, and financial strain, the precise identification and management of VTE risk factors are critical. Hence, there is an urgent need for standardized VTE prevention guidelines specifically tailored for Chinese MM patients to inform clinical practice [7].

Several models have been developed to predict VTE risk in MM patients, such as SAVED [8], IMPEDE-VTE [9], and PRISM [10,11]. However, these models exhibit varying levels of performance across different patient populations. In a cohort study of newly diagnosed MM patients in Brazil, researchers compared the SAVED and IMPEDE-VTE models. They reported that while the SAVED model performed poorly in predicting VTE risk, the IMPEDE-VTE model exhibited greater accuracy in risk classification. Although the SAVED score performed well in preliminary studies, it lacks extensive validation across independent populations [12]. Furthermore, the SAVED score is based on a limited number of risk factors, potentially overlooking significant factors [13]. Conversely, the IMPEDE-VTE score incorporates multiple variables, offering a more comprehensive risk assessment. While this improves predictive accuracy in specific contexts, it complicates clinical application. The NCCN guidelines recommend both the IMPEDE-VTE and SAVED models for VTE risk prediction in MM patients. However, it should be noted that although these two models have been studied in single-center cohorts in China [14], large-scale studies on the Chinese MM population are still limited, and the variables included in the existing models are not comprehensive enough. Furthermore, the PRISM score, proposed by Falanga et al., has demonstrated performance similar to or even worse than that of the SAVED and IMPEDE-VTE models in predicting VTE risk, particularly in assessing the impact of emerging therapies on VTE risk [15]. The PRISM score does not fully account for these novel factors, resulting in a decline in predictive accuracy. Consequently, significant limitations arise when these models are used to predict VTE risk in MM patients, primarily because they do not comprehensively encompass all VTE risk factors or account for the heterogeneity among MM patients [16].

Therefore, there is an urgent need to develop a VTE prediction model tailored explicitly for Chinese MM patients to enhance clinical practice and improve VTE prevention outcomes. A nomogram is a visual tool designed to predict the probability of clinical events in individual patients, converting traditional regression models into individualized visual risk assessments with increased practicality and precision. Previous studies have demonstrated the effectiveness of nomograms in predicting cancer prognosis, and they have been widely employed to assess VTE risk in patients with breast cancer, lymphoma, and ovarian cancer. However, research on VTE risk prediction in MM patients is still limited, and no appropriate predictive model has been established for this population. Accordingly, this study aims to develop a VTE prediction nomogram specifically tailored for MM patients to estimate their VTE risk more accurately. This model enables clinicians to identify high-risk VTE patients more precisely and implement timely preventive and therapeutic measures to reduce thrombosis risk, alleviate disease burden, and increase patient survival and quality of life [17]. Ultimately, this individualized risk assessment method based on a nomogram will bridge the gap in applying existing predictive models to Chinese MM patients, offering new directions for research and clinical practice in this field.

## 2. Materials and Methods

### 2.1. Study Population and Design

This study employed a retrospective cohort design to evaluate VTE risk in MM patients and develop an associated predictive model. The study’s data were obtained from Chongqing University Cancer Hospital and included MM patients treated between 2020 and 2022. A total of 845 patients were enrolled, each meeting predefined inclusion and exclusion criteria.

The inclusion criteria were as follows: (1) patients diagnosed with MM via bone marrow biopsy, imaging tests, and hematological examinations; (2) patients aged ≥18 years; and (3) patients hospitalized at least once at this institution and receiving standard treatment regimens, including but not limited to immunomodulatory drugs, proteasome inhibitors, and bone marrow transplantation. The exclusion criteria were as follows: (1) a prior history of VTE; (2) a diagnosis of other malignancies; (3) severely incomplete clinical data, follow-up records, or missing VTE event documentation; and (4) died within 48 h of admission.

This study was based on the retrospective collection of data following patients’ diagnoses, including detailed clinical information, treatment plans, follow-up records, and VTE event documentation. Potential VTE risk factors were selected based on the published literature, clinical experience, and relevant guidelines, incorporating a total of 25 variables. These variables included age, sex, body mass index (BMI), Karnofsky performance status (KPS) score, hypertension status, diabetes status, central venous pressure (CVP), international staging, radiotherapy, chemotherapy, targeted therapy, erythropoietin, dexamethasone, white blood cell count (WBC), D-dimers, and activated partial thromboplastin time (APTT), among others.

This study complies with the ethical principles of the Helsinki Declaration and received approval from the Ethics Committee of Chongqing University Cancer Hospital. All patient data were handled with strict confidentiality and used solely for scientific research purposes (Figure 1).

### 2.2. Statistical Analysis and Construction of the Nomogram

All data analyses for this study were conducted via R software (version 4.3.2). The following R packages were used for model development and evaluation: “car” (version 3.1-2), “survival” (version 3.5-7), “rms” (version 6.7-1), “tidyverse” (version 2.0.0), “skimr” (version 2.1.5), “foreign” (version 0.8-85), “timeROC” (version 0.4), “glmnet” (version 4.1-8), “MASS” (version 7.3-60), “ggDCA” (version 1.1), “pec” (version 2023.04.12), “riskRegression” (version 2023.12.21), “survIDINRI” (version 1.1-2), and “forplo” (version 0.2.5). An online nomogram server was developed via the “DynNom” package (version 5.1).

Participants were randomly assigned to the training and validation cohorts at a 7:3 ratio. In the training cohort, variables with less than 10% missing data were imputed via the “Mice” package (version 3.16.0). Univariate logistic regression analysis assessed the associations between candidate variables and VTE. Variables with a *p* value < 0.2 in the univariate analysis, along with those considered clinically significant, were included in the multivariate logistic regression analysis. Model calibration was assessed via the Hosmer–Lemeshow test and a nomogram was constructed based on the results of the multivariate analysis.

Descriptive statistics are presented as means ± standard deviations for continuous variables with a normal distribution, and differences between groups were analyzed via t-tests. The median and interquartile range (IQR) are reported for continuous variables not normally distributed, and nonparametric tests were used to compare group differences. Categorical data are reported as frequencies and percentages, and differences between groups were assessed via the chi-square test. The model’s classification performance in the training and validation cohorts was evaluated by plotting the receiver operating characteristic (ROC) curve and calculating the area under the curve (AUC). A calibration curve was used to evaluate the model’s predictive accuracy. The nomogram’s C-index and net benefit were used to evaluate its predictive ability and overall performance. Finally, clinical impact curves were plotted to assess the model’s efficacy in clinical applications.

## 3. Results

### 3.1. Clinical Characteristics of Patients

Between 2020 and 2022, 845 patients with MM were included in the analysis and randomly assigned to a training cohort (592, 70.06%) or a validation cohort (253, 29.94%) at a 7:3 ratio. In the training cohort, 343 patients were male, accounting for 57.94%, whereas 249 were female (42.06%). The patients had a mean age of 63.50 ± 11.02 years, and the average Karnofsky performance scale (KPS) score was 75.60. Among these patients, 413 (69.76%) were in stage III of tumor progression. Regarding treatment, most patients (442, 74.66%) received dexamethasone-based therapy, and 396 (66.89%) underwent adjuvant chemotherapy. A substantial proportion (434, 73.31%) opted against surgical intervention, while 53.72% received targeted therapy. A total of 148 patients with MM were diagnosed with VTE in this study, and the median time to VTE was 3.61 months after the start of treatment (Table 1).

### 3.2. Independent Predictive Factors in the Training Cohort

Univariate analysis revealed that patient age (*p* < 0.001), CVP (*p* < 0.001), and D-dimer level (*p* = 0.013) were significant risk factors for VTE. Patients who underwent surgery (*p* = 0.012) had a greater risk of developing VTE than those who did not. Furthermore, patients who received chemotherapy (*p* < 0.001), targeted therapy (*p* < 0.001), or anticoagulation therapy (*p* < 0.001) had a significantly greater risk of VTE. The use of erythropoietin and dexamethasone was also associated with an increased risk of VTE, which is consistent with the findings of previous studies. Laboratory indicators revealed that lymphocyte count (*p* = 0.039), Ca (*p* = 0.003), and APTT (*p* = 0.049) were protective factors against VTE.

Multivariate logistic regression analysis, incorporating variables deemed statistically significant and clinically relevant from the univariate analysis, identified age, KPS, surgery, chemotherapy, anticoagulation, erythropoietin use, and Hb, PLT, Ca, APTT, and D-dimer levels as independent predictors of VTE risk. Specifically, age was positively associated with the risk of VTE (OR = 1.03, 95% CI: 1.01–1.06, *p* = 0.015), i.e., the risk of VTE increased with each additional year of age. In addition, the risk of VTE increased with surgery (OR = 1.75, 95% CI: 1.04–2.94, *p* = 0.035), chemotherapy (OR = 2.16, 95% CI: 1.04–4.50, *p* = 0.040) and anticoagulation (OR = 4.14, 95% CI: 2.39–7.17, *p* < 0.001). Meanwhile, elevated levels of erythropoietin (OR = 2.56, 95% CI: 1.36–4.83, *p* = 0.004) and Hb (OR = 1.01, 95% CI: 1.01–1.02, *p* = 0.028) were associated with increased risk of VTE. In contrast, elevated levels of KPS score (OR = 0.97, 95% CI: 0.95–0.99, *p* = 0.002), PLT (OR = 0.99, 95% CI: 0.99–0.99, *p* = 0.019), Ca (OR = 0.31, 95% CI: 0.14–0.68, *p* = 0.040), and APTT (OR = 0.94, 95% CI: 0.89–0.99, *p* = 0.031) were protective factors, and elevated levels were associated with a reduced risk of VTE (Table 2).

### 3.3. Predictive Nomogram for VTE

Using the identified independent predictive factors, we developed a nomogram for predicting VTE risk (Figure 2). The nomogram provides a visual representation of the weighting of various variables in risk prediction and calibration according to the relationship between clinical data and expected outcomes.

In the nomogram, each risk factor is assigned a score on the upper scale, which is then summed on the lower total score scale to estimate the probability of VTE occurrence in MM patients. The nomogram calculates a total score for each MM patient by summing the scores of each risk variable. A higher total score indicates a greater risk of developing VTE. For example, a 64-year-old MM patient admitted to our hospital with a KPS score of 76, who had not undergone surgery, chemotherapy, anticoagulation, or received erythropoietin, had Hb, PLT, Ca, APTT, and D-dimer levels of 100, 101, 2, 28, and 2, respectively. According to the nomogram, this patient’s predicted risk of developing VTE was 5.10%. The nomogram revealed that APTT and calcium levels are the most significant contributors to VTE risk prediction, followed by platelet count, age, and Karnofsky performance scale (KPS). D-dimer and hemoglobin levels have a moderate impact on VTE risk.

Additionally, we developed an online calculator based on a nomogram (https://mmvte.shinyapps.io/DynNomapp/, accessed on 18 March 2025) (Figure 2). Unlike traditional risk assessment methods, the online calculator is distinguished by its simplicity of operation. Clinicians and other users need only input laboratory test results, clinical manifestations, and other relevant data into the designated fields of the calculator to promptly obtain the assessment results, thereby saving time and improving diagnostic and treatment efficiency. Additionally, the results generated by the online calculator can effectively mitigate subjective biases and provide a more objective and scientific basis for clinical decision-making.

### 3.4. Validation and Calibration of the Nomogram

For the nomogram for predicting VTE risk in MM patients, the C-index was 0.811 (95% CI: 0.766–0.855) in the training cohort and 0.714 (95% CI: 0.626–0.801) in the validation cohort, as shown in Figure 3A,B. These findings indicate that the nomogram has a superior discriminative ability for predicting VTE risk. The calibration curves for both the training cohort (Figure 3C) and the validation cohort (Figure 3D) exhibited low Brier scores and smooth curves, reflecting strong agreement between the observed values and the predicted probabilities.

### 3.5. Evaluating the Effectiveness of the Nomogram in Clinical Decision-Making

This study integrates decision curve analysis (DCA) and the clinical impact curve (CIC) to evaluate the clinical applicability of the developed prediction model comprehensively. Initially, DCA was utilized to assess the net benefit of the model across different thresholds, aiding in the determination of the optimal threshold for clinical decision-making. Subsequently, the CIC was applied to examine the predictive impact at the specified threshold.

As illustrated in Figure 4A,B, the optimal threshold ranges were 10–63% for the training cohort and 20–45% for the validation cohort. Within these ranges, the nomogram outperformed both the “treat all” and “treat none” strategies, offering superior clinical benefits for patients with MM. The results of the DCA suggest that the nomogram performs effectively within this threshold range, demonstrating high clinical utility and facilitating informed decision-making. The CIC demonstrates the number of patients predicted to be at high risk and recommended for intervention and those likely to benefit from the intervention at various thresholds. For example, as shown in Figure 4C,D, at a 20% threshold, the model identified approximately 340 high-risk patients, approximately 160 of whom were expected to require intervention and benefit from it. At the 60% threshold, the model’s predictions closely matched the actual outcomes, reflecting high predictive accuracy in clinical practice.

## 4. Discussion

VTEs are of significant clinical relevance in patients with malignant tumors. VTE not only increases morbidity and mortality, but also impacts the selection and efficacy of cancer treatments [18]. In the context of MM, VTE poses significant challenges, making accurate VTE risk assessment and individualized thromboprophylaxis essential components of supportive care [4]. Several VTE risk prediction tools have been developed and implemented for MM patients, including the Khorana score, the SAVED model, the IMPEDE VTE model, and the PRISM score [12,19]. Although these models have shown strong performance in predicting VTE risk among MM patients, they have notable limitations, including insufficient sample size for inclusion and a lack of consideration of relevant factors [8].

As visual predictive tools in oncology, nomograms are pivotal in advancing personalized medicine [20]. Numerous studies have utilized nomogram models for predicting VTE risk in cancers such as colorectal cancer, breast cancer, non-small-cell lung cancer, and lymphoma [21,22,23,24]. However, there is limited research on nomogram-based VTE risk prediction models for MM patients. In response, we developed a nomogram model tailored to the Chinese population to predict VTE risk in MM patients. This model integrates various complex risk factors and generates individualized risk scores, facilitating the accurate prediction of each patient’s risk of VTE. The nomogram enhances its readability and practical utility by converting complex regression equations into straightforward, intuitive graphics. In addition, recent studies have demonstrated that other malignant tumors, including neuroendocrine tumors, exhibit numerous physiological and pathological similarities to MM-associated VTE [5,25]. This discovery introduces a novel perspective for cross-disciplinary research, suggesting that the MM-associated VTE risk model developed in this study could be a foundation for predicting VTE risk in other malignant tumors. Furthermore, it offers a valuable reference for enhancing thrombosis prevention and treatment across the field of oncology.

In this study, we developed a nomogram incorporating age, KPS, surgery, chemotherapy, anticoagulation, erythropoietin use, and APTT, Hb, PLT, Ca, and D-dimer levels. The C-index of 0.811 indicates substantial predictive accuracy, and the model has been internally validated. Multivariate logistic regression analysis revealed that age, surgery, chemotherapy, and anticoagulation were significant risk factors for VTE in MM patients. Previous studies have demonstrated that the incidence of VTE significantly increases in patients aged over 65 years [26]. Additionally, the use of chemotherapeutic agents such as thalidomide and lenalidomide is strongly associated with VTE because of their propensity to activate platelets and increase coagulation tendencies, thereby increasing the risk of thrombosis [27]. Li et al. were the first to apply the IMPEDE-VTE score to Chinese MM patients, achieving an AUC of 0.72 [14]. In contrast, our nomogram demonstrated superior predictive accuracy. Although studies have shown an increased risk of VTE in MM patients receiving dexamethasone, especially those with a history of high-dose treatment [28], dexamethasone did not emerge as a significant risk factor in our study. This may be due to the influence of anticoagulation therapy, which could have mitigated the VTE risk associated with dexamethasone. Furthermore, our study revealed that incorporating D-dimer level into the nomogram improved predictive performance, in line with the findings by Sanfilippo et al., who reported that D-dimer level enhanced the predictive power of the IMPEDE-VTE score in newly diagnosed MM patients after chemotherapy [29]. VTE risk in MM patients exhibits significant heterogeneity [30], reflecting patient variability due to individual characteristics, disease status, treatment modalities, and biomarkers [31]. To address this heterogeneity, we developed a web-based nomogram, in addition to the model, enabling clinicians to make personalized predictions and select tailored treatment strategies for various patient profiles. Finally, it should be noted that the present study found that indicators such as platelet count, D-dimer level, and anticoagulation are likely to be common predictors of the risk of VTE associated with multiple malignancies [32]. This finding is significant and could provide a valuable reference for subsequent research when conducting cross-tumor studies on predicting VTE risk.

However, several limitations should be acknowledged in this study. First, the data for this study were obtained from patients with MM at the Chongqing University Cancer Hospital, and this specific population and region may limit the nomogram’s applicability. Second, additional external validation is needed to increase the robustness of the model. Third, while the variables included were chosen based on the literature and clinical experience, some potentially influential factors, such as lifestyle habits, were not considered. Future studies should incorporate a broader range of biomarkers and clinical variables, as well as multicenter data to validate the generalizability of the model.

## 5. Conclusions

This study developed and validated a nomogram model for predicting the risk of VTE in patients with MM. The model demonstrated high predictive accuracy by integrating a range of clinical and laboratory variables, which was further substantiated through internal validation. Additionally, we created a user-friendly web-based tool to aid clinicians in providing individualized VTE risk assessments and preventive strategies. Although the findings of this study are promising, the generalizability of the model is constrained by the use of single-center data. Further multicenter external validation studies are needed to establish its broader applicability. Future research should also incorporate additional biomarkers and clinical factors to enhance the model’s comprehensiveness and predictive power, thereby advancing the application of personalized medicine in managing patients with MM.

## Figures and Tables

**Figure 1 biomedicines-13-00770-f001:**
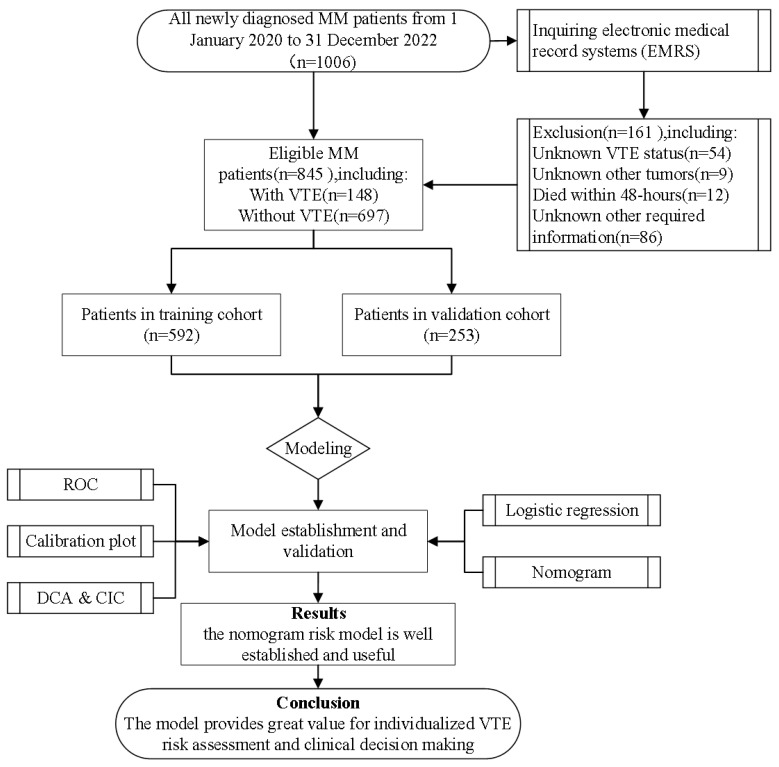
Flowchart of the VTE risk modeling process for MM patients. MM: multiple myeloma; ROC: receiver operating characteristic; DCA: decision curve analysis; CIC: clinical impact curve.

**Figure 2 biomedicines-13-00770-f002:**
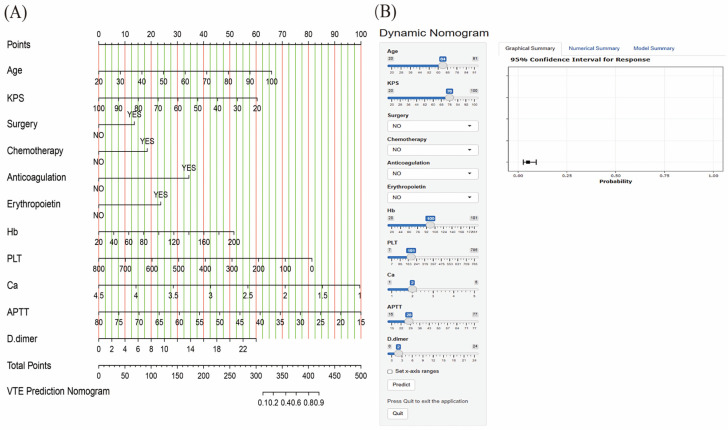
VTE prediction nomogram for MM patients (**A**) and the interface of the web-based nomogram (**B**).

**Figure 3 biomedicines-13-00770-f003:**
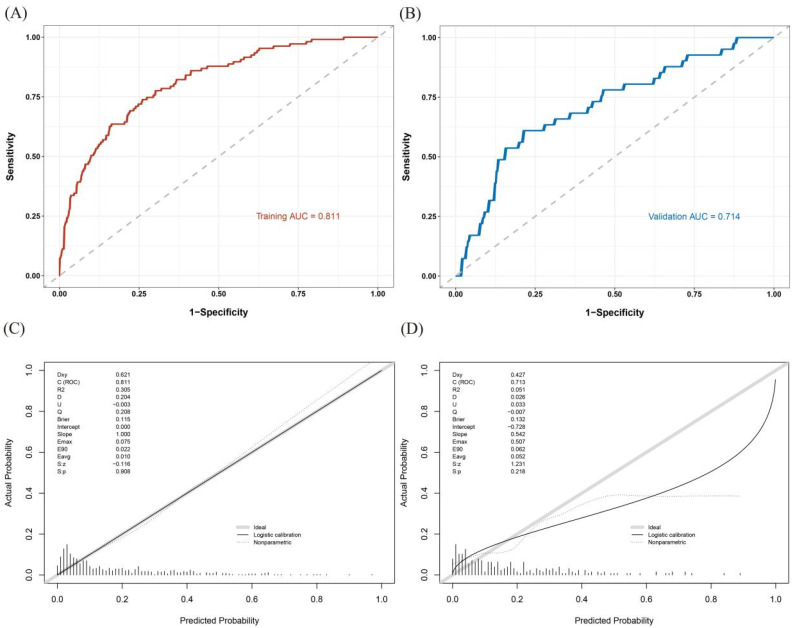
ROC curves for predicting VTE in the training cohort (**A**) and validation cohort (**B**). Calibration plots of the VTE risk nomogram in the training cohort (**C**) and validation cohort (**D**).

**Figure 4 biomedicines-13-00770-f004:**
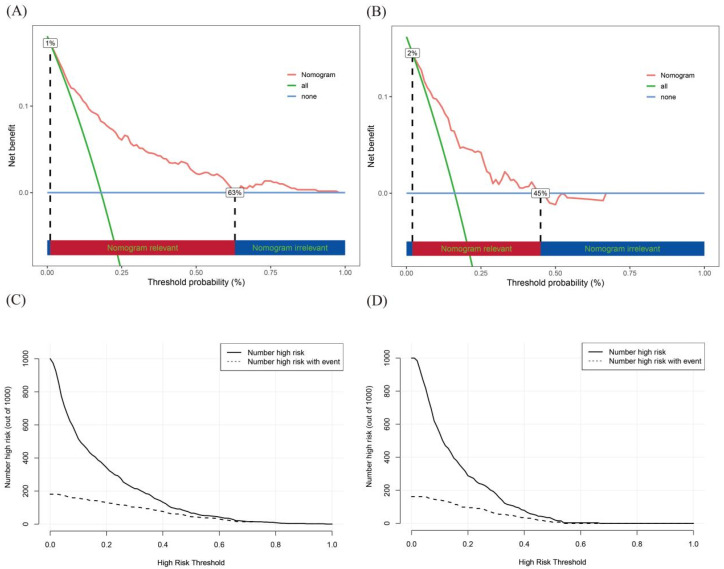
DCA curves of VTE in the training cohort (**A**) and DCA curves of VTE in the validation cohort (**B**); clinical impact curves (CICs) for VTE in the training cohort (**C**) and the validation cohort (**D**).

**Table 1 biomedicines-13-00770-t001:** Demographic and clinicopathological characteristics of the training and validation cohorts.

Variables	Overall (*n* = 845)	Training Cohort (*n* = 592)	Validation Cohort (*n* = 253)	*p*-Value
Age (year)	63.53 ± 11.01	63.50 ± 11.02	63.61 ± 11.01	0.890
KPS (points)	75.74 ± 11.58	75.60 ± 11.78	76.06 ± 11.10	0.597
BMI				
≤28	791 (93.61)	555 (93.75)	236 (93.28)	0.919
>28	54 (6.39)	37 (6.25)	17 (6.72)	
Sex				
Male	492 (58.22)	343 (57.94)	149 (58.89)	0.856
Female	353 (41.78)	249 (42.06)	104 (41.11)	
Hypertension				
NO	647 (76.57)	458 (77.36)	189 (74.70)	0.455
YES	198 (23.43)	134 (22.64)	64 (25.30)	
Diabetes				
NO	739 (87.46)	523 (88.34)	216 (85.38)	0.280
YES	106 (12.54)	69 (11.66)	37 (14.62)	
CVP				
NO	541 (64.02)	383 (64.70)	158 (62.45)	0.586
YES	304 (35.98)	209 (35.30)	95 (37.55)	
Fracture				
NO	678 (80.24)	483 (81.59)	195 (77.08)	0.157
YES	167 (19.76)	109 (18.41)	58 (22.92)	
Paralysis				
NO	736 (87.10)	514 (86.82)	222 (87.75)	0.799
YES	109 (12.90)	78 (13.18)	31 (12.25)	
ISS				
I–II	264 (31.24)	179 (30.24)	85 (33.60)	0.377
III	581 (68.76)	413 (69.76)	168 (66.40)	
Surgery				
NO	625 (73.96)	434 (73.31)	191 (75.49)	0.564
YES	220 (26.04)	158 (26.69)	62 (24.51)	
Chemotherapy				
NO	289 (34.20)	196 (33.11)	93 (36.76)	0.344
YES	556 (65.80)	396 (66.89)	160 (63.24)	
Targeted				
NO	386 (45.68)	274 (46.28)	112 (44.27)	0.643
YES	459 (54.32)	318 (53.72)	141 (55.73)	
Anticoagulation				
NO	523 (61.89)	370 (62.50)	153 (60.47)	0.633
YES	322 (38.11)	222 (37.50)	100 (39.53)	
Erythropoietin				
NO	734 (86.86)	517 (87.33)	217 (85.77)	0.614
YES	111 (13.14)	75 (12.67)	36 (14.23)	
Dexamethasone				
NO	214 (25.33)	150 (25.34)	64 (25.30)	1.000
YES	631 (74.67)	442 (74.66)	189 (74.70)	
VTE				
NO	697 (82.49)	485 (81.93)	212 (83.79)	0.578
YES	148 (17.51)	107 (18.07)	41 (16.21)	
WBC (10^9^/L) *	5.43 [4.16, 6.98]	5.46 [4.16, 6.99]	5.24 [4.16, 6.78]	0.621
Hb (g/L)	100.12 ± 27.91	99.61 ± 27.29	101.30 ± 29.34	0.419
PLT (10^9^/L) *	176.00 [131.00, 239.00]	176.00 [128.00, 240.00]	176.00 [136.00, 233.00]	0.845
LYM (10^9^/L) *	1.32 [1.02, 1.74]	1.33 [1.01, 1.73]	1.31 [1.03, 1.76]	0.940
β2.Microglobulin (mg/L)*	4.50 [3.00, 7.60]	4.48 [3.00, 7.53]	4.70 [3.00, 7.60]	0.637
Ca (mmol/L)	2.34 ± 0.38	2.32 ± 0.38	2.36 ± 0.40	0.231
APTT (s)	28.20 ± 5.71	28.31 ± 5.88	27.94 ± 5.31	0.390
PT (s)	12.07 ± 1.89	12.10 ± 1.84	12.00 ± 2.03	0.484
D.dimer (mg/L) *	0.85 [0.37, 1.92]	0.84 [0.38, 1.91]	0.91 [0.33, 1.99]	0.967

* expressed as the median [P25, P75]. KPS: Karnofsky performance status; BMI: body mass index; CVP: central venous pressure; ISS: International Staging System; Hb: hemoglobin; PLT: platelet; LYM: lymphocyte; APTT: activated partial thromboplastin time; PT: prothrombin time; WBC: white blood cell; Ca: calcium.

**Table 2 biomedicines-13-00770-t002:** Univariate and multivariate analyses of the risk factors for VTE.

Characteristics	No VTE (*n* = 485)	VTE (*n* = 107)	OR (Univariable)	OR (Multivariable)
Age (year)	62.65 ± 10.83	67.36 ± 11.09	1.04 (1.02–1.07, *p* < 0.001)	1.03 (1.01–1.06, *p* = 0.015)
KPS (points)	76.42 ± 10.89	71.87 ± 14.67	0.97 (0.96–0.99, *p* < 0.001)	0.97 (0.95–0.99, *p* = 0.002)
BMI				
≤28	452 (93.20)	103 (96.26)		
>28	33 (6.80)	4 (3.74)	0.53 (0.18–1.53, *p* = 0.243)	
Sex				
Male	275 (56.70)	68 (63.55)		
Female	210 (43.30)	39 (36.45)	0.75 (0.49–1.16, *p* = 0.195)	
Hypertension				
NO	376 (77.53)	82 (76.64)		
YES	109 (22.47)	25 (23.36)	1.05 (0.64–1.73, *p* = 0.842)	
Diabetes				
NO	429 (88.45)	94 (87.85)		
YES	56 (11.55)	13 (12.15)	1.06 (0.56–2.02, *p* = 0.860)	
CVP				
NO	330 (68.04)	53 (49.53)		
YES	155 (31.96)	54 (50.47)	2.17 (1.42–3.32, *p* < 0.001)	
Fracture				
NO	398 (82.06)	85 (79.44)		
YES	87 (17.94)	22 (20.56)	1.18 (0.70–2.00, *p* = 0.527)	
Paralysis				
NO	424 (87.42)	90 (84.11)		
YES	61 (12.58)	17 (15.89)	1.31 (0.73–2.35, *p* = 0.361)	
ISS				
I–II	150 (30.93)	29 (27.10)		
III	335 (69.07)	78 (72.90)	1.20 (0.75–1.92, *p* = 0.436)	
Surgery				
NO	366 (75.46)	68 (63.55)		
YES	119 (24.54)	39 (36.45)	1.76 (1.13–2.75, *p* = 0.012)	1.75 (1.04–2.94, *p* = 0.035)
Chemotherapy				
NO	182 (37.53)	14 (13.08)		
YES	303 (62.47)	93 (86.92)	3.99 (2.21–7.21, *p* < 0.001)	2.16 (1.04–4.50, *p* = 0.040)
Targeted				
NO	244 (50.31)	30 (28.04)		
YES	241 (49.69)	77 (71.96)	2.60 (1.64–4.11, *p* < 0.001)	
Anticoagulation				
NO	336 (69.28)	34 (31.78)		
YES	149 (30.72)	73 (68.22)	4.84 (3.09–7.60, *p* < 0.001)	4.14 (2.39–7.17, *p* < 0.001)
Erythropoietin				
NO	435 (89.69)	82 (76.64)		
YES	50 (10.31)	25 (23.36)	2.65 (1.55–4.53, *p* < 0.001)	2.56 (1.36–4.83, *p* = 0.004)
Dexamethasone				
NO	140 (28.87)	10 (9.35)		
YES	345 (71.13)	97 (90.65)	3.94 (1.99–7.77, *p* < 0.001)	
WBC (10^9^/L) *	5.55 [4.20, 6.99]	5.15 [3.96, 6.99]	0.99 (0.93–1.05, *p* = 0.782)	
Hb (g/L)	99.32 ± 27.83	100.91 ± 24.78	1.00 (0.99–1.01, *p* = 0.586)	1.01 (1.01–1.02, *p* = 0.028)
PLT (10^9^/L) *	180.00 [127.00, 240.00]	160.00 [131.50, 222.00]	0.99 (0.99–1.00, *p* = 0.173)	0.99 (0.99–0.99, *p* = 0.019)
LYM (10^9^/L) *	1.36 [1.06, 1.75]	1.09 [0.88, 1.56]	0.69 (0.49–0.98, *p* = 0.039)	
β2.Microglobulin (mg/L) *	4.50 [3.03, 7.80]	4.00 [2.80, 6.60]	1.01 (0.98–1.03, *p* = 0.619)	
Ca (mmol/L)	2.35 ± 0.39	2.23 ± 0.33	0.35 (0.18–0.71, *p* = 0.003)	0.31 (0.14–0.68, *p* = 0.004)
APTT (s)	28.53 ± 6.18	27.30 ± 4.10	0.96 (0.91–1.00, *p* = 0.049)	0.94 (0.89–0.99, *p* = 0.031)
PT (s)	12.11 ± 1.90	12.02 ± 1.55	0.97 (0.86–1.10, *p* = 0.638)	
D.dimer (mg/L) *	0.78 [0.35, 1.78]	1.25 [0.63, 3.12]	1.08 (1.02–1.14, *p* = 0.013)	1.11 (1.03–1.20, *p* = 0.006)

* expressed as the median [P25, P75]. KPS: Karnofsky performance status; BMI: body mass index; CVP: central venous pressure; ISS: International Staging System; Hb: hemoglobin; PLT: platelet; LYM: lymphocyte; APTT: activated partial thromboplastin time; PT: prothrombin time; WBC: white blood cell; Ca: calcium.

## Data Availability

The datasets generated and/or analyzed during the current study are not publicly available due to local legal requirement but are available from the corresponding author on reasonable request.

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
