# Peer review of "Development and Validation of a Predictive Nomogram for Venous Thromboembolism Risk in Multiple Myeloma Patients: A Single-Center Cohort Study in China"

_biomedicines, 2025, doi:10.3390/biomedicines13040770_

Round 1

Reviewer 1 Report

Comments and Suggestions for Authors

Title ” Development and validation of a predictive nomogram for ve-2 nous thromboembolism risk in multiple myeloma patients: A single center cohort study in China ”

Please add in the title ” A single-center cohort study ”

Pg. 3 Line 10-11”MM patients with VTE face a 3x.....” needs a reference to sustain this affirmation.

Pg. 3 Line 20-21. Please note the appropriate reference next to each score and cite the original articles for each of these scores. For example is not mentioned the original work for the IMPEDE VTE score:  Sanfilippo KM, Luo S, Wang TF, Fiala M, Schoen M, Wildes TM, Mikhael J, Kuderer NM, Calverley DC, Keller J, Thomas T, Carson KR, Gage BF. Predicting venous thromboembolism in multiple myeloma: development and validation of the IMPEDE VTE score. Am J Hematol. 2019 Nov;94(11):1176-1184. doi: 10.1002/ajh.25603.

Note that the IMWG score is diagnostic score, not a VTE risk evaluation. I think it should not be mentioned as a VTE prediction score.

 Pg. 3 Line 32-34: ”however, neither has undergone extensive validation in Asian populations, limiting their applicability to Chinese MM patients.”  This sentence is not accurate because there are studies that have validated these scores on the Chinese population (reference 13 and 25). I think it should be reformulated. This is not the reason that limits applicability.

Figure 1, V  is written ”Eligible MM patients (n=845 ),including: With VTE(n=697) Without VTE(n=148)”. I think is a mistake, a data inversion.

Please add explanations in for all abbreviations used in the tables or figures (BMI, CVP and so on)

Material and methods: Explain which population is being treated at your center. Is this representative of the general population of China or your region? This clarification is important because we cannot generalize and claim that this model is applicable to the entire Chinese population.

Explain why there are differences between the results presented in the text (pg 7 line 13-19 and pg 8 line 1-5  and the data in Table 2.

Although a criterion for choosing variables for multivariate analysis is a p below 0.2 in univariate analysis, you did not include CVP, targeted immunotherapy, dexamethasone, and LYM in the multivariate analysis, although, as Table 2 shows, they should have been included. Please explain why.

Author Response

Comments1: Title“Development and validation of a predictive nomogram for ve-2 nous thromboembolism risk in multiple myeloma patients: A single center cohort study in China ”.

Response 1: Thanks for the valuable comments. This is an excellent suggestion, and we have added “single-center cohort study” to the title.

Comments2: Pg. 3 Line 10-11”MM patients with VTE face a 3x.....” needs a reference to sustain this affirmation.

Response 2: We sincerely appreciate your valuable comments. We have carefully reviewed the literature and added references supporting the statement “Compared with patients without VTE, patients with MM who have VTE have a three-fold increased mortality rate within one year” in the relevant sections of the revised manuscript (DOI:10.1016/j.thromres.2021.08.015.)

Comments3: Pg. 3 Line 20-21. Please note the appropriate reference next to each score and cite the original articles for each of these scores. For example is not mentioned the original work for the IMPEDE VTE score:  Sanfilippo KM, Luo S, Wang TF, Fiala M, Schoen M, Wildes TM, Mikhael J, Kuderer NM, Calverley DC, Keller J, Thomas T, Carson KR, Gage BF. Predicting venous thromboembolism in multiple myeloma: development and validation of the IMPEDE VTE score. Am J Hematol. 2019 Nov;94(11):1176-1184. doi: 10.1002/ajh.25603.

Note that the IMWG score is diagnostic score, not a VTE risk evaluation. I think it should not be mentioned as a VTE prediction score.

Response 3: We sincerely thank the reviewer for the careful reading. Following the reviewer's suggestion, we have added the referenced literature next to each risk assessment model in the revised manuscript. We are very grateful for the reviewer's comment that “the IMWG score is a diagnostic score rather than a risk assessment.” We have removed the relevant statement in the manuscript.

Comments4: Pg. 3 Line 32-34: ”however, neither has undergone extensive validation in Asian populations, limiting their applicability to Chinese MM patients.”  This sentence is not accurate because there are studies that have validated these scores on the Chinese population (reference 13 and 25). I think it should be reformulated. This is not the reason that limits applicability.

Response4: Thanks for your keen academic insight. We have revised the relevant wording. The statements in lines 32–34 on page 3 of the original manuscript have been adjusted to: “However, it should be noted that although both models have been evaluated in single-center cohort studies in China, large-sample studies targeting the Chinese MM population are still relatively rare, and there are issues such as insufficiently comprehensive variables included. Therefore, there is an urgent need to construct risk prediction models with large samples and multiple variables to improve the applicability and predictive accuracy of the models.”

Comments5:  Figure 1, V  is written ”Eligible MM patients (n=845 ),including: With VTE(n=697) Without VTE(n=148)”. I think is a mistake, a data inversion.

Response 5: Thanks for your valuable comments. Due to an oversight in the manuscript preparation process, there was confusion between the number of MM patients with VTE and those without VTE in Figure 1. We have corrected this error in the revised manuscript.

Comments6: Please add explanations in for all abbreviations used in the tables or figures (BMI, CVP and so on).

Response 6: Thanks for carefully reading our manuscript. We have provided corresponding supplementary explanations for the abbreviations in the tables and images based on your suggestions to improve the article's readability and comprehension. Thanks again for your valuable feedback.

Comments7: Material and methods: Explain which population is being treated at your center. Is this representative of the general population of China or your region? This clarification is important because we cannot generalize and claim that this model is applicable to the entire Chinese population.

Response 7: Thanks for your valuable comments on our manuscript. During the manuscript preparation, we failed to adequately describe the treatment population and the model's applicability in the Materials and Methods section. We have now added supplementary information in the Discussion section: “The data for this study were derived from patients with multiple myeloma at the Chongqing University Cancer Hospital. The applicability of the findings may be limited to the specific population and region.” Once again, we appreciate your meticulous review and valuable suggestions.

Comments8: Explain why there are differences between the results presented in the text (pg 7 line 13-19 and pg 8 line 1-5  and the data in Table 2.

Response 8: Thanks for your careful reading and for pointing out the errors in our manuscript. Upon re-examination, we realized that due to our oversight, there were mistakes in the content and explanation of Table 2. We have now provided a detailed re-explanation of the data in Table 2. Once again, we appreciate your thorough review.

Comments9: Although a criterion for choosing variables for multivariate analysis is a p below 0.2 in univariate analysis, you did not include CVP, targeted immunotherapy, dexamethasone, and LYM in the multivariate analysis, although, as Table 2 shows, they should have been included. Please explain why.

Response 9: Thanks for your valuable comments. In the univariate analysis, we set the variable selection criterion to P < 0.2 to avoid excluding potentially important variables. The variables selected in the univariate analysis include Age, KPS, sex, CVP, Surgery, Chemotherapy, Targeted, Anticoagulation, Erythropoietin, Dexamethasone, PLT, LYM, Ca, APTT, and D.dimer.

Based on the selected variables, we conducted multivariate analysis again. Multivariate analysis aims to assess each variable's independent effects while controlling for other variables.  Therefore, a stricter P-value criterion (P < 0.05) is required to control the false-positive rate. The final variables selected in the multivariate analysis are Age, KPS, Surgery, Chemotherapy, Anticoagulation, Erythropoietin, PLT, Ca, APTT, and D.dimer. It should be noted that Table 2 presents the final significant variables after multivariate selection.

Additionally, in addition to using statistical methods for variable selection, we also incorporated clinical expertise and findings from previous studies. We forced the inclusion of specific indicators for the final multivariate analysis. For example, although Hb did not show statistical significance in the univariate analysis, we still included it in the multivariate analysis, which ultimately showed statistical significance.

Reviewer 2 Report

Comments and Suggestions for Authors

This study addresses a clinically significant issue by developing and validating a nomogram to predict venous thromboembolism (VTE) risk in multiple myeloma (MM) patients. Given the high morbidity and mortality associated with VTE in MM, predictive models are crucial for risk stratification and clinical decision-making. The study is well-structured, employs rigorous statistical methods, and presents a practical clinical tool with an online calculator for individualized VTE risk assessment.

However, a few aspects require improvement:

    • The manuscript focuses exclusively on MM but does not contextualize its findings within the broader spectrum of cancer-associated thrombosis (CAT).
    • Neuroendocrine tumors (NETs) are another malignancy where VTE risk is often underestimated, yet recent research (DOI: 10.3390/cancers17020212) suggests shared pathophysiological mechanisms between NET-related thrombosis and MM-associated VTE, including coagulation abnormalities, tumor-secreted prothrombotic factors, and treatment-related thrombotic risks. Including a discussion of VTE in other malignancies (such as NETs) would broaden the clinical applicability of the predictive model and highlight potential similarities or differences in risk factors across hematologic and solid tumors.
      • The study provides internal validation of the nomogram, but it lacks external validation in an independent cohort.
      • A brief discussion on the need for prospective multicenter validation would further strengthen the manuscript.
      • The authors highlight the potential clinical use of the online nomogram, but they should provide a discussion on how this model compares with existing risk stratification tools, such as the IMPEDE-VTE or SAVED models.
        • Can this model be adapted for other hematologic malignancies (e.g., lymphomas) or even solid tumors like NETs?
        • The inclusion of the reference (DOI: 10.3390/cancers17020212) would help reinforce the idea that cancer-associated thrombosis is not exclusive to MM, and similar predictive approaches could be applied to other malignancies.
        • Are there shared predictors of VTE between MM and other cancers, such as platelet count, D-dimer levels, or coagulation biomarkers
        • Would the nomogram be applicable to other malignancies, or is it MM-specific?
        • The addition of external validation would improve model robustness.
        • Would integrating biomolecular markers (e.g., tissue factor expression, microparticles, or genetic thrombophilia) enhance predictive power?
        • Could a similar model be developed for other cancers, like NETs, which are also associated with unexpected VTE events?

Minor Comments

  • The manuscript is well-written but could benefit from minor language polishing to improve clarity and conciseness.

    • The nomogram and calibration plots are well-presented, but a brief explanation in the legend on how to interpret the nomogram would be helpful for readers unfamiliar with this tool.
      •  

Author Response

Comments1: The manuscript focuses exclusively on MM but does not contextualize its findings within the broader spectrum of cancer-associated thrombosis (CAT).

Response 1: Thanks for your valuable suggestions. Since our research group focuses on multiple myeloma, we would first like to establish a VTE risk assessment model specifically for MM patients. Once the effectiveness of this model has been widely validated in clinical practice, we plan to collaborate with researchers from other groups to conduct more extensive studies on cancer-related thrombosis.

Comments2: Neuroendocrine tumors (NETs) are another malignancy where VTE risk is often underestimated, yet recent research (DOI: 10.3390/cancers17020212) suggests shared pathophysiological mechanisms between NET-related thrombosis and MM-associated VTE, including coagulation abnormalities, tumor-secreted prothrombotic factors, and treatment-related thrombotic risks. Including a discussion of VTE in other malignancies (such as NETs) would broaden the clinical applicability of the predictive model and highlight potential similarities or differences in risk factors across hematologic and solid tumors.

Response 2: Thanks for your valuable comments. Our team reviewed the literature and found that neuroendocrine tumors (NETs) and multiple myeloma (MM) have a substantial similarity in VTE risk. We have added relevant discussions in the paper and supplemented the common pathological mechanisms mentioned in the relevant literature (DOI: 10.1002/Cam4.1927, DOI: 10.3390/cancers17020212). In the future, our team plans to apply the model to neuroendocrine tumors to test its applicability.

Comments3: The study provides internal validation of the nomogram, but it lacks external validation in an independent cohort.

Response 3: Thanks for the valuable comments provided by the reviewer. External validation is indeed crucial for model reliability and generalization ability. However, due to limitations in data sources, our study has not yet undergone external validation. We have explained the limitations of the paper and suggested that in future research, the model's universality should be validated based on a broader patient population or multi-center data. In addition, our team has contacted the heads of multiple centers to discuss the initial draft of the multi-center research. In the future, we will collaborate with multiple centers to conduct relevant.

Comments4: A brief discussion on the need for prospective multicenter validation would further strengthen the manuscript.

Response 4: Thanks for the valuable suggestions from the reviewer. We fully agree with the role of prospective multi-center validation in enhancing the reliability and generalizability of research results. We have explained the limitations of the paper and suggested that in future research, the model's universality should be validated based on a broader patient population or multi-center data. In addition, our team has contacted the heads of multiple centers to discuss the initial draft of the multi-center research. We will collaborate with multiple centers to conduct relevant research in the future. Thanks again for your valuable feedback and suggestions.

Comments5: The authors highlight the potential clinical use of the online nomogram, but they should provide a discussion on how this model compares with existing risk stratification tools, such as the IMPEDE-VTE or SAVED models.

Response 5: Thanks for your valuable suggestion. We fully agree on comparing our online nomogram with existing VTE risk stratification tools such as IMPEDE-VTE and SAVED models. This comparison not only helps to accurately evaluate our model's clinical advantages and innovation but also provides a more valuable reference for clinical practice.

However, while trying to implement this suggestion, we encountered some difficulties. Despite our efforts to contact relevant research authors using IMPEDE-VTE and SAVED models to obtain specific parameters and validation methods for the models, unfortunately, our attempts have not been successful so far.

Due to the lack of this key information, we cannot directly compare these three models based on existing data, which is very regrettable. To address this deficiency, we have added an overview of existing VTE risk stratification tools in the discussion section and emphasized the unique features of our model in terms of data sources, predictor variable selection, and clinical applicability. These supplements can provide readers with more comprehensive background information and help them better understand the potential advantages of our model.

Comments6: Can this model be adapted for other hematologic malignancies (e.g., lymphomas) or even solid tumors like NETs?

Response 6: Thanks for your question. Your question about the model's applicability is crucial and provides us with further direction for thinking and exploration. Although the model in this study was developed for patients with multiple myeloma, its potential applicability may be extended to other hematological malignancies (such as lymphoma) and certain solid tumors (such as neuroendocrine tumors). The construction of this model is based on the integration of multivariate data, covering clinical features, laboratory indicators, and treatment-related factors. These factors may also have similar effects in other hematological malignancies, especially when the pathological features and treatment background are similar. For example, lymphoma patients may also face similar thrombotic risk factors when receiving immunomodulators or chemotherapy, making this model's theoretical framework somewhat universal. To validate this hypothesis, our team plans to apply our model to other hematological malignancies and solid tumors in future studies to test its applicability and accuracy. We hope to further expand the application scope of the model through these studies and provide support for thrombus risk assessment in more cancer patients.

Comments7: The inclusion of the reference (DOI: 10.3390/cancers17020212) would help reinforce the idea that cancer-associated thrombosis is not exclusive to MM, and similar predictive approaches could be applied to other malignancies.

Response 7: Thanks for your valuable suggestions. We fully agree with you that cancer-related thrombosis is not limited to multiple myeloma (MM) and that similar predictive methods apply to other malignancies. We have included in the manuscript the literature you mentioned (DOI: 10.3390/cancers17020212) to support this view further.

Comments8: Are there shared predictors of VTE between MM and other cancers, such as platelet count, D-dimer levels, or coagulation biomarkers.

Response 8: Thanks for raising fundamental questions. We searched relevant literature and found (DOI: 10.1002/cam4.7231) that some hematological and coagulation indicators, such as platelet count, D-dimer levels, and coagulation factors, may be common predictors of VTE risk in various malignant tumors. Our study mainly focused on the risk of VTE in patients with multiple myeloma. However, we recognize that these indicators may also play a similar role in other cancers. We have added a discussion on this issue in the manuscript.

Comments9: Would the nomogram be applicable to other malignancies, or is it MM-specific?

Response 9: Thanks for the questions raised by the reviewer. The nomogram we are currently developing is based on multiple myeloma patients' specific clinical characteristics and VTE risk factors. Therefore, its design and construction are mainly aimed at patients with multiple myeloma findings. However, we also recognize that specific VTE predictors, such as platelet count, D-dimer levels, and coagulation biomarkers, may have similar predictive effects in other malignant tumors. Therefore, although the current nomogram is specific to MM, we believe some of its key predictive factors may also apply to other malignant tumors, especially those cancer types with similar pathological mechanisms, such as hypercoagulability and tumor-associated thrombosis risk. To further evaluate its universality, we will conduct cross-tumor type studies in the future to validate whether the nomogram applies to other malignant tumors or whether adjustments are needed based on the characteristics of different tumors. Thanks again for your suggestions and opinions.

Comments10: The addition of external validation would improve model robustness.

Response 10: Thanks for the valuable suggestions provided by the reviewer. We fully agree that external validation is crucial for improving the model's robustness and clinical application value. Our research is based on single-center queue data, and the model has only undergone internal validation. Due to practical limitations such as external data acquisition and resource issues, we are temporarily unable to conduct external validation, which also constitutes one of the limitations of our research. Our team has contacted the leaders of multiple centers to discuss the initial draft of the multi-center research. We will collaborate with multiple centers to conduct relevant research in the future.

Comments11: Would integrating biomolecular markers (e.g., tissue factor expression, microparticles, or genetic thrombophilia) enhance predictive power?

Response 11: Thanks for the valuable comments. Our current research focuses on the impact of clinical-pathological factors and treatment plans on the risk of VTE. Although biomolecular markers such as tissue factor expression, microparticles, and hereditary thrombophilia may enhance predictive ability, these factors have not yet been included due to our research design and data sources. In future research, we plan to explore further how to incorporate these biomarkers into VTE prediction models to enhance their predictive ability.

Comments12: Could a similar model be developed for other cancers, like NETs, which are also associated with unexpected VTE events?

Response 12: Thanks for the question. We believe that a similar model can be constructed if some tumors are also associated with sudden venous thromboembolism events. However, the model needs to be appropriately adjusted due to differences in the biological characteristics, treatment plans, and risk factors associated with VTE among different tumors. For example, regarding NETs mentioned by the reviewer, we may need to adjust variable selection based on their unique biomarkers or treatment factors while also considering the epidemiological differences in VTE occurrence among NET patient populations. Thanks for your suggestion. In future multi-center studies, we will design relevant indicators for validation.

Comments13: The manuscript is well-written but could benefit from minor language polishing to improve clarity and conciseness.

Response 13: Thanks for your affirmation and valuable suggestions on our manuscript. We highly value your feedback and have invited a professional English author to carefully polish the entire text to enhance its readability and conciseness of expression. We believe that these improvements can better present our research content.

Comments14: The nomogram and calibration plots are well-presented, but a brief explanation in the legend on how to interpret the nomogram would be helpful for readers unfamiliar with this tool.

Response14: Thanks for their positive feedback on the nomogram and calibration chart. We have provided examples of interpreting column lines on page 9, lines 11-16, and page 10, lines 1-6 of the manuscript. However, in order to further assist readers who are not familiar with this tool in understanding, we briefly added some explanations when constructing the nomogram in the manuscript, such as "a nomogram is used to show the weights of different variables in risk prediction, and calibrated through the relationship between clinical data and expected events." This brief prompt helps improve the readability and understanding of the chart.

Reviewer 3 Report

Comments and Suggestions for Authors

Development and validation of a predictive nomogram for venous thromboembolism risk in multiple myeloma patients: A

cohort study in China

The study addresses a significant health issue in multiple myeloma patients and aims to provide a practical tool for physicians to enhance medical care.

The authors  use of a retrospective cohort design with logistic regression analysis and validation techniques such as ROC curves and C-index enhances the credibility of the results. Which ara suitable statistical methods

Comparing the nomogram with other models like IMPEDE-VTE and SAVED which provides a clear perspective on the efficiency and effectiveness of the newly developed model.

 The development of an online calculator makes the research more applicable in clinical practice, give the study practice view.

Points need to be reviewed

The conclusion need to be summarize

The figures need to be more clear for reader

Larger sample may give more validity of resutls

Add more vairable as lifestyle habits, nutrition, and family history of thrombosis could enhance the model’s predictive accuracy. May support study results

Long follow up is recommended

Author Response

Comments1: The study addresses a significant health issue in multiple myeloma patients and aims to provide a practical tool for physicians to enhance medical care.

Response 1: Thanks for their affirmation of our research. We are well aware of the health challenges patients with multiple myeloma face. This study aims to provide clinical doctors with a practical predictive tool to help optimize treatment plans and improve patients' quality of life and prognosis.

Comments2: The authors  use of a retrospective cohort design with logistic regression analysis and validation techniques such as ROC curves and C-index enhances the credibility of the results. Which ara suitable statistical methods.

Response 2: Thanks for recognizing our research design and statistical methods.

Comments3: Comparing the nomogram with other models like IMPEDE-VTE and SAVED which provides a clear perspective on the efficiency and effectiveness of the newly developed model.

Response 3: Thanks for your valuable suggestion. We fully agree on the importance of comparing our online nomogram with existing VTE risk stratification tools such as IMPEDE-VTE and SAVED models. This comparison not only helps to accurately evaluate our model's clinical advantages and innovation but also provides a more valuable reference for clinical practice.

However, in trying to implement this suggestion, we encountered some difficulties. Despite our efforts to contact relevant research authors using IMPEDE-VTE and SAVED models to obtain specific parameters and validation methods for the models, our attempts have not been successful so far.

Due to the lack of this key information, we cannot directly compare these three models based on existing data, which is very regrettable. To address this deficiency, we have added an overview of existing VTE risk stratification tools in the discussion section and emphasized the unique features of our model in terms of data sources, predictor variable selection, and clinical applicability. These supplements can provide readers with more comprehensive background information and help them better understand the potential advantages of our model.

Comments4: The development of an online calculator makes the research more applicable in clinical practice, give the study practice view.

Response 4: Thanks for the valuable feedback. To make research more applicable to clinical practice, we will supplement relevant practical perspectives in the paper on how to improve the efficiency and accuracy of clinical work through the development of online calculators. Thanks for the guidance. Your feedback has been constructive in improving our research.

Comments5: The conclusion need to be summarize.

Response 5: Thanks for the valuable feedback. We have re-examined the conclusion section and added a concise summary to highlight the main findings and contributions of the research. We have refined the conclusion section to make it more general and clearly articulate the research's core conclusions and practical significance. Thanks again for your feedback.

Comments6: The figures need to be more clear for reader.

Response 6: Thanks for the valuable comments. We have rechecked the charts in the paper and modified their annotations and explanations to make them more intuitive and easy to understand.

Comment7: Larger sample may give more validity of resutls.

Response 7: Thanks for the valuable suggestions. We fully agree that increasing the sample size helps to improve the reliability and statistical power of research results. Although our current sample size has been reasonably planned during the research design phase based on expected outcomes and research objectives, we also realize that a larger sample size can further enhance the results' broad applicability and statistical robustness. We have provided additional explanations in the discussion section and plan to expand the sample size further for multicenter prospective studies in future research. Thanks again for your feedback.

Comments8: Add more vairable as lifestyle habits, nutrition, and family history of thrombosis could enhance the model's predictive accuracy. May support study results.

Response8: Thanks for the valuable suggestions. This study mainly focused on clinical and laboratory indicators in existing datasets. However, we also realized that variables such as lifestyle habits, nutritional status, and family history of thrombosis can improve predictive accuracy and enhance their clinical applicability. We have added this content in the discussion section. Thanks again for your suggestion, which is of great significance for us in improving our relevant research in the future.

Comments9: Long follow up is recommended.

Response9: Thanks for the valuable suggestions provided by the reviewer. We fully agree that long-term follow-up can further validate the stability and predictive accuracy of the model and help capture long-term trends and potential risk factors. Therefore, our team will continue to track and follow up on all study subjects in this study further to validate the accuracy and clinical applicability of the model. Thanks again for your suggestion.

Round 2

Reviewer 1 Report

Comments and Suggestions for Authors

I appreciate all the improvements made by the authors. 

Please add explanations for the abbreviations used in Figure 1

Author Response

Comments 1: I appreciate all the improvements made by the authors. Please add explanations for the abbreviations used in Figure 1.

Response: Thanks a lot for your professional suggestions. We've modified Figure 1 in the manuscript and added explanations for the abbreviations.

Reviewer 2 Report

Comments and Suggestions for Authors

The authors made all changes required. 

Author Response

Comments 1: The authors made all changes required. 

Response: Thank you very much for your professional suggestions, which enable our research results for better presentation.